# Towards Efficient and Accurate Identification of Memorization in Deep Models

## Abstract

*Memorization* is the ability of deep models to learn verbatim arbitrary inputs from the training data. One of the most popular means of calculating memorization scores (i.e., the probability that a point is memorized) is via the pseudo Leave-One-Out (pLOO) method proposed by Feldman & Zhang (2020). However, this technique suffers from two shortcomings: it is computationally prohibitive (as it requires training thousands of models) and it produces inaccurate scores. The goal of this work is to overcome both these limitations simultaneously. To do so, we take the following approach: **First**, we demonstrate that the major source of pLOO's computation bottleneck is its execution on the entire dataset, not just the memorized points. We find running pLOO on all the points is unnecessary since most of them are not even memorized. **Second**, we develop a simple proxy to identify the memorized points without having to run pLOO in the first place. To do so, we study the model training cycle and find that memorized points are learned towards the last iterations. We build a simple proxy based on this observation and find that our proxy: *a)* is strongly correlated with the actual memorization scores (Pearson score $< -0.95$) across all our models and datasets and *b)* requires only a single model (instead of the thousands needed by pLOO). However, our proxy does not provide the exact memorization scores. **Third**, to calculate these, we incorporate our proxy into the pLOO method, resulting in pLOO$_{improved}$. In doing so, we show that our pLOO$_{improved}$ reduces both computational overhead (by over 90%) and the error in the approximated memorization scores (by over 65%). Therefore, our work makes it possible to study memorization in large datasets and real-world models while requiring only a fraction of the computational resources.

## 1 Introduction

**What is memorization:** Machine Learning (ML) models have the propensity to memorize arbitrary train-label pairs (Zhang et al., 2017). Points that are most vulnerable to memorization usually belong to small sub-populations (Abdullah et al., 2023b; Feldman, 2020). These include outliers (e.g., a single black cat in a dataset of white cats), malformed inputs (e.g., a blurry or garbled cat image), mislabeled points (e.g., a cat mislabeled as an ORANGE), etc. This is because small sub-populations have few overlapping features with the remaining points in the label set (i.e., outliers, malformed points, and mislabeled inputs do not look like the other images in the label set). As a result, the features learned from the rest of the label distribution are not useful for classifying the small sub-populations. Consequently, the model is forced to rely on memorization to correctly classify these points. As a result, identifying memorized points is important for tasks such as detecting privacy leakage, fairness, data mislabeling etc (Black & Fredrikson, 2021; Usynin et al., 2024; Carlini et al., 2022b; 2023; 2021; Somepalli et al., 2023; Ye et al., 2022; Salem et al., 2018; Zarifzadeh et al., 2024; Jayaraman et al., 2020; Shokri et al., 2017; Long et al., 2020).

**How to find memorized points:** The gold standard method to predict whether a point $x$ is memorized is via a Leave-One-Out (LOO) test (Section 3.2) (Feldman, 2020). Here, we train two models, one on the full dataset and another with the point $x$ removed. If the two models classify $x$ with different labels (i.e., the model correctly classifies the point only when it is present in the data), then the point is likely memorized. However, if the models output similar predictions (i.e., both models correctly classify the point whether or not it is present in the data) then the point is likely *generalized*.

However, testing every point individually for memorization via LOO is prohibitively expensive as it requires training a separate pair of models for each point in the training data. To add to the computational cost, LOO must be repeated dozens of times for each point to account for various sources of randomness (Feldman & Zhang, 2020). As a result, executing LOO on a small dataset of 50,000 points (like CIFAR-10) will require training hundreds of thousands of models, a computationally expensive endeavor.

**The current approximation method:** To overcome this bottleneck, Feldman & Zhang (2020) proposed a pseudo-Leave-One-Out (pLOO) procedure (Section 5). Instead of removing a single point at a time, the authors randomly drop a fixed percentage of the data and train a *shadow* model on the resulting data shard. The authors repeat this procedure thousands of times so that every point is evenly distributed across the data shards. For example, if the user creates 2,000 data shards, each consisting of a random sample of 50% of the original data, then each point is present in approximately half the shards. To check if a point is memorized, the authors calculate the *memorization score*. This refers to the difference in the number of models that classified the point correctly when the point was *included* versus when the point was *excluded*. Points with the highest memorization scores (i.e., ones that are only classified correctly when they are present in the data) are marked as memorized, while the ones with the lowest scores are considered generalized (Abdullah et al., 2023b). This approximation method reduces the number of required models from hundreds of thousands to just a few thousand. As a result, this approximation method became a basis of numerous attacks and defenses in the space of ML privacy (Salem et al., 2018; Zarifzadeh et al., 2024; Carlini et al., 2022a; Watson et al., 2021; Long et al., 2020; Sablayrolles et al., 2019; Song & Mittal, 2021; Jayaraman et al., 2020; Shokri et al., 2017; Yeom et al., 2018; Tang et al., 2022).

**Limitation of the approximation method:** However, the pLOO method has two major limitations. First, it still requires training thousands of shadow models, which is not practical for either large datasets consisting of hundreds of thousands of points (e.g., Imagenet) or large production models that comprise billions of parameters. Second, as we show later in our work (Section 5.2), pLOO over-estimates the memorization scores (Root Mean Square Error of 35.5). This is because pLOO creates data shards by dropping large chunks of data (often tens of thousands of points), unlike the baseline LOO method, which only removes a single point at a time. As a result, pLOO's memorization scores are often an overestimation of the actual LOO baseline, which can inevitably lead researchers to draw incorrect conclusions. To overcome these severe shortcomings, our goal is to develop a method that is both **more efficient** and **more accurate** than pLOO.

**How we do this:** The pLOO procedure calculates the memorization score for *each* point in the dataset since it does not have any a priori information about the memorized points. We show that this step is computationally wasteful since only a fraction of the dataset is memorized (Section 4.1). Indeed, a more efficient method should only evaluate the memorized points, instead of the full training data (which contains both memorized and generalized points). However, it is not clear from existing literature how to differentiate between the generalized and memorized points without having to run pLOO in the first place. To answer this question, we study the model training cycle (Section 4.2). We hypothesize that:

*Generalized points are learned faster than memorized ones during training.*

We empirically validate this hypothesis across several standard architectures (MobileNet, VGG19, RESNET18) and data sets (CIFAR-10, CIFAR-100, and Tiny ImageNet) and use it to develop a simple proxy to identify points that are most likely memorized (Section 4.3). We develop a proxy the memorization scores that consists of training a single model on the full data set and calculating the Accuracy per Batch (ApB) i.e., the percentage of batches for which a point is classified correctly. Our ApB proxy is strongly correlated with the actual memorization scores (Pearson score $< -0.95$ across all datasets and models). The negative correlation means points with the lowest ApB have high memorization scores. As a result, we can use our proxy to reduce the pLOO search space, from the entire dataset to just the points with the lowest ApB score (Section 5.1).

While our proxy can identify the memorized points, it does not provide their *exact* memorization scores. The availability of these scores might be necessary based on the specific use case, such as for infering membership inference (Carlini et al., 2022a; Zarifzadeh et al., 2024). To calculate these, we incorporate our proxy into the pLOO method, resulting in pLOO*improved*. Specifically, pLOO*improved* creates data shards from *only* the points with the lowest ApB score, instead of the entire dataset.

This reduces the number of pLOO shards by over 90% (from a few thousand to just a few hundred) and consequently, reduces the number of shadow models.

Even though our method is significantly faster than pLOO, it does *not* come at the cost of memorization score accuracy. When compared to the LOO baseline, the original pLOO has a much higher Root Mean Squared Error (RMSE) of 35.5. On the other hand, pLOO$_{improved}$ reduces the error to 12.19, a reduction of 65% (Section 5.2). **In short, our method is both significantly *faster* and *more accurate* than pLOO**, the most popular method for finding memorized points in current literature. Our work makes the following contributions:

1. We develop a simple proxy to identify memorized points. Our proxy has a large negative correlation ($< -0.95$) with memorization scores across *all* the models and datasets we evaluate, indicating a strong relationship between the two.

2. We develop pLOO$_{improved}$ by incorporating our proxy into pLOO to reduce its search space.

3. As a result, pLOO$_{improved}$ not only reduces the computational overhead (by over 90%) but also improves memorization score accuracy (by over 65%).

## 2    INTUITION BEHIND PROXY

Our improved method is based on the hypothesis that generalized points are learned faster than memorized ones. Let's take a moment to consider why we believe our hypothesis will be true. Recall that how fast the model learns a point depends on the gradient of the training batch. Broadly speaking, two aspects can impact this per batch gradient: sub-population size (Zielinski et al., 2020) and its loss (Goodfellow et al., 2016).

1. **Sub-population:** Gradients from the same sub-populations point in the same direction (Neyshabur et al., 2017). One can imagine each image in the batch having a small individual gradient pointing in some direction. Images with similar features (i.e., ones belonging to the same sub-population) will have gradients that point in a similar direction. Since the final gradient is the sum of the per-sample gradient, the larger the sub-population, the larger their combined impact on the final direction (Zielinski et al., 2020).

2. **Loss:** The higher the loss, the larger the gradients (Goodfellow et al., 2016). This is because the training method updates the model weights more aggressively to minimize the loss. However, as training continues and loss decreases, so does the size of the gradients.

During the first few training epochs, most points are misclassified and have an equally high loss. However, large sub-populations, due to their size, will have a greater aggregate loss and therefore, a higher contribution towards the per batch gradient direction (Chatterjee, 2020; Zielinski et al., 2020). The model will adjust its weights accordingly, learning the larger sub-populations in the earlier epochs (Paul et al., 2021). Since large sub-populations usually consist of generalized points (i.e., low memorization score) (Feldman & Zhang, 2020), the generalized will be simultaneously learned.

As training progresses, the large sub-populations will continue to be learned and consequently, their loss will slowly decrease. After enough training epochs, their aggregate loss will decrease and will eventually be close to that of the small sub-populations. At this point, the gradients of the small sub-populations will start to impact the per-batch gradients as well. The model will adjust its weights to reduce the loss over the small sub-populations. As a consequence, the model will learn the memorized points because they usually constitute the small sub-populations. While the problem has been studied from the lens of artificial memorization (i.e., noisy input and noisy labels) Stephenson et al. (2021), it has yet to be verified for natural points. Since artificial and natural points can behave in a diverging ways Aerni et al. (2024), part of our contribution to is to show that natural points behave in the same manner.

# 3 IDENTIFYING MEMORIZED POINTS

## 3.1 MEMORIZATION DEFINITION

According to the Feldman & Zhang (2020), a point is considered memorized if the model's output label changes when the point is removed from the dataset. The memorization score is the percentage of the models that assigned the ground-truth label when it was *inside* in the training set minus the percentage of models that assigned the ground-truth label when the point *outside* the training set:

$$\mathbf{Pr}_{h \leftarrow A(S)}[h(x_i) = y_i] - \mathbf{Pr}_{h \leftarrow A(S^{\setminus i})}[h(x_i) = y_i] \tag{1}$$

Here, $x_i$ is a point in the training set $S$ where $S = ((x_1, y_1)...(x_n, y_n))$, $(h \leftarrow A(S))$ are models($h$) trained on the full data ($S$) using an algorithm ($A$) and ($h \leftarrow A(S^{\setminus i})$) are models trained on data after removing point $x_i$ ($S^{\setminus i}$).

In other words, *if* we train 2000 instances of each of the models (i.e., 1000 models where $x_i$ is present and 1000 models where $x_i$ absent from the training data ). 100% models output the ground-truth label for $x_i$ (i.e., all 1000 instances classified the point correctly). However, once $x_i$ is removed from the training data, 25% of the models assign $x_i$ the ground-truth label (i.e., 250 out of the 1000 instances classified the point correctly). The resulting memorization score is $100\% - 25\% = 75\%$. In contrast, if there is an insignificant change in the memorization score (i.e., $x_i$ is classified correctly, whether or not it is present in the training data) then we do not mark point $x_i$ as memorized.

Furthermore, if the memorization score is close to 100%, then it was only classified correctly when present in the training data. Therefore, this point belongs to a sub-population of size one (i.e., it is an outlier). If the score is closer to 0, then the point was classified correctly even if it was absent from the training data. This means that the point belongs to a large sub-population consisting of many points. In general, the lower the score, the larger the sub-population, and the larger the score, the smaller the sub-population (Abdullah et al., 2023a).

## 3.2 LEAVE-ONE-OUT (LOO)

So far, we have only defined the memorization score in Equation 1. One way to calculate this value is via the classic LOO: 1) Train a model on the full data 2) Remove one point. 3) Retrain the model on the remaining data 4) Repeat steps 1, 2, and 3 dozens of times to account for the different sources of randomness (e.g., the varying initialization, GPU randomness, etc.). 5) Calculate the memorization score. 6) Repeat 1-5 for each point in the dataset. This methodology requires us to train hundreds of thousands of models, which is computationally intractable for even small datasets.

## 3.3 PSEUDO LEAVE-ONE-OUT (PLOO)

To overcome this limitation, Feldman & Zhang (2020) propose a method to *approximate* the memorization scores using a three-step process. Step 1: Instead of removing a single point, they sample a fraction $r$ from the *entire* data set (originally of size $n$), where $0 < r < 1$. Step 2: The sampled fraction or data shard is then used to train the model. In Feldman & Zhang (2020), the authors use $r = 0.7$ and repeat it $k$ times. The exact value of $k$ depends on the dataset but is typically on the order of a few thousand models. As a result, a random point $x_i$ is present in approximately $k \cdot r$ models and is absent from $k \cdot (1 - r)$. Step 3: Use Equation 1 to approximate the memorization score for each point in the data set. The pLOO method reduces the number of models needed from a few hundred thousand (needed by LOO) to a few thousand. However, training even this number of models is not possible for most production models (that require weeks or months to train) or researchers (who do not belong to well-funded research labs with large GPU clusters).

Considering these limitations, the goal of our work is to further reduce the number of required models to make it computationally tractable for researchers to study memorization on real-world datasets and models.

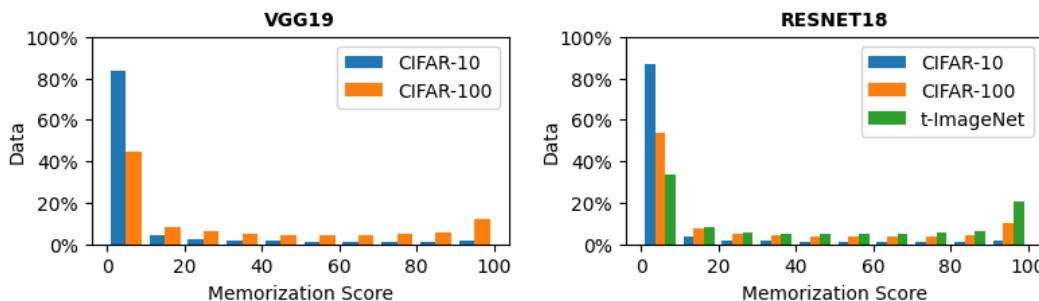

Figure 1: The figure above shows the histogram of the memorization scores. We can see that in all cases, most of the data has lower memorization scores (i.e., it is generalized) while a much smaller percentage of the data has high memorization score (i.e., it is memorized). Results for MobileNet and Resnet50 in Figure 6, in the Appendix.

## 4 DEVELOPING A PROXY

In this Section, we will develop a simple proxy to identify the memorized points without having to run the expensive pLOO procedure in the first place. We do so by answering some fundamental questions: **1) What portion of the data is memorized?** We find that models usually memorize only a fraction of the dataset. However, pLOO calculates the memorization score for *every* point in the set, both generalized and memorized. As a result, running pLOO on the entire dataset is computationally wasteful, since we only require the scores for the memorized points. Unfortunately, pLOO does not have a priori knowledge of which points are memorized, thereby motivating the need to build a proxy. Next, we take the first step towards building this proxy by testing our original hypothesis i.e., **2) Are generalized points learned sooner than memorized ones?** Here, we empirically evaluate and validate our hypothesis across different models and datasets. Having done so, we use our hypothesis to answer the question, **3) Can we build a simple proxy to identify the memorized points?** We develop the ApB proxy that consists of counting the number of batches for which a point is classified correctly. We show that ApB is strongly correlated to the memorization scores, and therefore, can be used to identify memorized points.

Our experimental setup is similar to prior works (Feldman & Zhang, 2020; Abdullah et al., 2023a). To account for architecture/dataset variety, we employ three different models (MobileNet (Howard et al., 2017), VGG19 (Simonyan & Zisserman, 2014), and RESNET18 (He et al., 2016)) across three different datasets (CIFAR-10, CIFAR-100 (Krizhevsky et al., 2009), and TinyImageNet (Le & Yang, 2015)). We run pLOO by training 2000 models for 100 epochs, using a batch size of 512, with a triangular learning rate of 0.4, and weight decay of $5e^{-4}$. We use the FFCV library (Leclerc et al., 2023) to improve the training speed and Random Horizontal Flip and Random Translate (padding=2). We use Equation 1 to calculate the memorization score for each point in the dataset. In total, we train around 14,000 models across 10 V100 GPUs over a few weeks. While we evaluate all three models on the CIFAR datasets, due to the computational load of training thousands of models, we evaluate TinyImageNet (t-ImageNet) only on RESNET18. Unless explicitly stated, all Mobilenet experiments are in the Appendix. Having described our setup, we can now answer the initial questions.

### 4.1 WHAT PORTION OF THE DATA IS MEMORIZED?

Figure 1 shows the results of our experiments using the original pLOO method. As a reminder, generalized points have lower memorization scores while memorized ones have a higher score (Section 3.1). We can see across all our experiments in Figure 1, a small fraction of the data is memorized. The degree of memorization also seems to be tied to the number of output labels. For example, in the case the RESNET18, we can see that CIFAR-10 (10 labels) has the fewest memorized points, followed by CIFAR-100 (100 labels) and t-ImageNet (200 labels). Similarly, CIFAR-10 has consistently fewer memorized points across all models compared to CIFAR-100. This is possibly due to the train-test gap (Leino & Fredrikson, 2020; Salem et al., 2018; Yeom et al., 2018), where the

larger the train-test gap, the greater the memorization. While the degree of memorization might vary across datasets and architectures, our results show that **models memorize only a fraction of the data.** As a result, running pLOO on the entire dataset is unnecessary. Furthermore, even in the extreme case where the models memorize most of the data (which is true for models with low test accuracy), running the vanilla pLOO algorithm is still computationally wasteful. As we do not want to spend compute calculating scores of points that the model generalized By only focusing on the memorized ones, we can significantly reduce the pLOO search space and the resulting computational costs. However, what is not yet clear is how to find these memorized points without having to run pLOO in the first place.

## 4.2 TESTING THE HYPOTHESIS: ARE GENERALIZED POINTS LEARNED SOONER THAN MEMORIZED ONES?

Now, we examine the training cycle to evaluate our hypothesis. We believe that generalized samples will be learned sooner than memorized ones. Since generalized points consist of large subpopulations, we believe generalized points will be learned in the earlier epochs while the memorized points will be learned towards the end (Section 2). To ascertain if this is the case, we calculate the classification accuracy of the memorized and generalized points at each epoch. To account for catastrophic forgetting due to different sources of stochasticity during training and weight initialization (Jagielski et al., 2022; Tirumala et al., 2022; Toneva et al., 2018; Graves et al., 2021; Toneva et al., 2018), we repeat this experiment 50 times on the full dataset and aggregate the classification accuracy of each point across all the models (i.e., a point is considered learned if it is classified correctly across all 50 models).

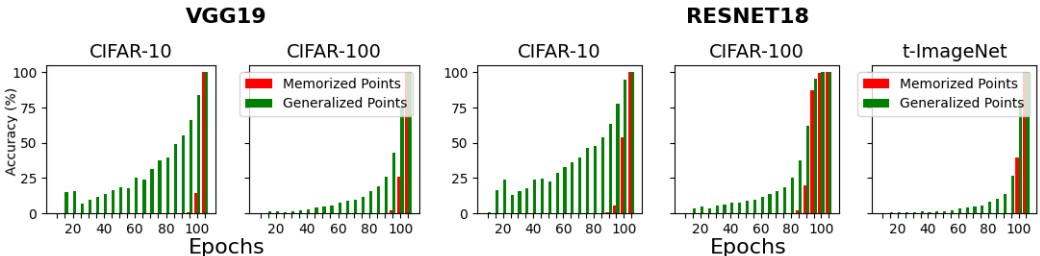

Figure 2: The figure shows the empirical validation of our hypothesis (Section 2). We can see that in all cases, the generalized points (green) are learned earlier in the training cycle, compared to the memorized ones (red). Results for MobileNet and Resnet50 in Figure 7, in the Appendix.

The results are shown in Figure 2. We can see that the generalized points (green) are learned right from the start. Their classification accuracy begins increasing from the initial epochs. On the other hand, the accuracy of the memorized points (red) remains zero until much later in the training cycle. These observations are consistent across all the datasets and architectures that we evaluated. These results provide an experimental validation for our hypothesis and **generalized points are learned sooner than the memorized ones.**

## 4.3 CAN WE BUILD A SIMPLE PROXY TO IDENTIFY THE MEMORIZED POINTS?

Having found evidence for our hypothesis, we can now build a simple proxy to differentiate the memorized points from the generalized ones. Our metric consists of counting the number of batches for which a point was classified correctly or Accuracy per Batch (ApB) (shown in Algorithm 1). This simple metric provides a few advantages: 1) It accounts for our hypothesis. This is because the earlier the point is learned in the training cycle, the higher the ApB. 2) It addresses catastrophic forgetting during training, namely, points that are initially learned and then forgotten have a lower ApB. 3) It is easy to compute. 4) Most importantly, it requires training only a single model.

To evaluate our proxy, we train one model on the entire dataset and calculate the ApB. Since calculating ApB for every single batch can be expensive (as it requires running inference on the entire

---

**Algorithm 1** Accuracy per Batch (ApB): Our proxy only requires adding 4 lines of code (1,4, 6, and 7) to the standard training loop.

---

**Require:** Dataset $D = \{(x_i, y_i)\}$, Model $M$, Epochs $E$, Batch size $B$
**Ensure:** ApB scores ApB($x_i$) for all $x_i \in D$
 1: Initialize ApB($x_i$) $\leftarrow 0 \; \forall x_i \in D$
 2: **for** $e = 1$ to $E$ **do**
 3:     **for** each batch $B_j = \{(x_i, y_i)\}$ **do**         ▷ For every batch or every Nth batch
 4:         $\hat{Y}_j \leftarrow M(X_j)$         ▷ Model predictions for batch
 5:         **for** each $(x_i, y_i) \in B_j$ **do**
 6:             **if** $\hat{y}_i = y_i$ **then** ApB($x_i$) $\leftarrow$ ApB($x_i$) $+ 1$
 7:         Update $M$ using $B_j$         ▷ Backward pass
 8: **return** ApB($x_i$) $\forall x_i \in D$

---

training dataset), we calculate the ApB over the last batch of each epoch. We repeat this process 50 times to ascertain whether we get consistent results.

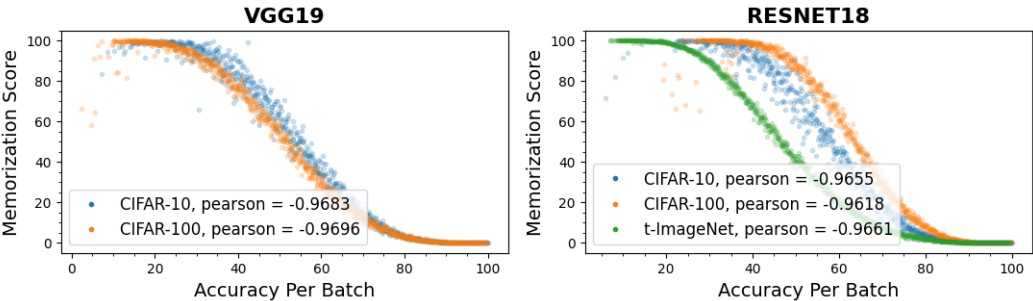

Figure 3: Figure shows the relation between ApB and memorization scores (calculated via pLOO). Points with the highest memorization scores have the lowest ApB. Therefore, the ApB proxy can help identify the memorized points without having to run the expensive pLOO procedure. Results for MobileNet and Resnet50 in Figure 8, in the Appendix.

---

Figure 3 shows the ApB values of each point versus its memorization score (calculated via pLOO) and the corresponding correlation scores (in the legend). We can see that in all cases, there is a strong negative Pearson correlation ($< -0.95$) between our proxy and the memorization scores. Visually, we can see that there is a clear relationship between the two: points with high memorization scores have a low ApB value. Points with close to zero memorization scores have very high ApB values, while ones with close to the maximum memorization score have very low ApB scores. We see that this trend is consistent across all models and datasets. Therefore, a point with a low ApB is likely memorized. As a result, **ApB proxy can identify memorized points using only a single model.** However, our proxy does not provide the *exact* memorization scores.

## 5 IMPROVING PLOO

To calculate the scores, we incorporate the proxy into the original pLOO method. The resulting method, pLOO$_{improved}$, reduces the total number of data shards (i.e., data partitions consisting of random subset of points) and consequently, the number of required shadow models. As a result, we reduce the computational overhead of calculating the memorization scores by 90% and reduce the error in the estimated scores by 65%.

### 5.1 REDUCE TOTAL DATA SHARDS

As mentioned in Section 3, pLOO creates shards from the entire data set by randomly sampling a fixed percentage of the points. Each shard is then used to train a shadow model, resulting in a

training bottleneck. Since pLOO is creating shards from the entire data set, it is effectively sampling from both the generalized and memorized points. This is unnecessary since most of the points are not memorized in the first place (Section 4.1).

To fix this issue, the user can employ the ApB proxy to first identify the generalized (highest ApB scores) and memorized points (lowest ApB scores). When creating the data shards, the user should only sample from the memorized points. As a result, each data shard will contain all the generalized points but only a sub-sample of the memorized ones. Concretely, consider a dataset $D$ of size $s$ that contains $m$ memorized points and $g$ generalized ones, where $s = m + g$. When creating data shards, a user has a sampling fraction $r$ (i.e., the fraction of points to select randomly). As a result, the size of each shard will be $g + r \cdot m$.

Next, we need to calculate how many data shards we need. In the original paper, the authors create 2000 data shards when sampling from a dataset of 50,000 points, which is approximately 0.04 shards per point (Feldman & Zhang, 2020; Abdullah et al., 2023b;a). We use the same ratio to calculate the number of required data shards. For example, if we are evaluating 5,000 memorized points, then we need $5,000 \cdot 0.04 = 200$ shards. As a result of the smaller sample space, we require fewer data shards, and consequently, fewer shadow models.

## 5.2 SETUP:

We perform our evaluation across all the models and datasets described earlier in Section 5. We train a single model to extract the ApB scores using the same recipe in Section 5 and select the top 5,000[1] points as memorized. Next, we train 200 models (as opposed to the 2,000 needed by the original pLOO method). We use a sample ratio $r$ of 50% so we have an even distribution of points across the 200 data shards using the sharding algorithm outlined in the previous subsection. Our evaluation consists of two parts:

**Part 1:** We compare the memorization scores between our pLOO*improved*, and the original pLOO method. We calculate the difference in memorization scores using the RMSE between each point's score from both of the methods. We run this evaluation across datasets and model architectures outlined in the previous section. This reveals how the scores vary across different training scenarios.

**Part 2:** Next, we evaluate which of the two methods is more accurate (i.e., closer to baseline LOO procedure). However, since LOO is computationally prohibitive to execute on every point in the dataset (Section 3.2), we run it over 150 points that had the largest difference in memorization scores between the original pLOO and pLOO*improved*, using a VGG-6 architecture trained on CIFAR-10. We choose this setup for three reasons. 1) VGG-6 model has a high training speed (2 mins per model). As a result, we can train the additional 3,200 models (20 models per point for a total of 160 points) required for the baseline LOO experiment. This is just not possible for the other models we evaluate in this work due to their slow training speeds. 2) The VGG-6 model has a high accuracy (Train Set: 99%. Test Set: 88%), indicating high utility. 3) pLOO and LOO are model-independent methods (i.e., not tied to any specific architecture). Therefore, we have no reason to believe that our findings will not extend to other models.

## 5.3 RESULTS AND DISCUSSION:

**Part 1: pLOO*improved* vs pLOO:** Figure 4 shows the RMSE difference between the two methods. Our results demonstrate that:

1. The more complex the dataset, the smaller the difference in scores. We can explain this phenomenon by observing the link between the number of memorized points from Figure 1 and RMSE in Figure 4. Specifically, the larger the number of points with a high memorization score, the smaller the RMSE difference. For example, RESNET-18 on t-ImageNet has the largest number of points with a high memorization score, but the lowest RMSE in Figure 4. This is because we are only evaluating the *top* 5,000 points with the highest score, most of which have the same maximum memorization score (close to 100%) for t-ImageNet. There-

---

[1]Since there is a linear relationship between the ApB values and the memorization scores (Section 4.3), most of the actual memorized points are within this range. While this is adequate for our evaluation, a user might choose more or fewer points based on their ML task and the computational resources.

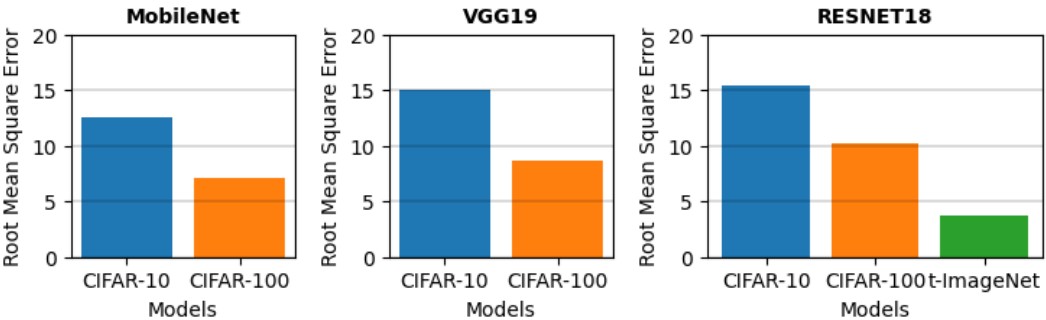

Figure 4: The difference between memorization scores calculated using the original pLOO and our pLOO$_{improved}$, quantified using the RMSE. Even though there is a difference between two scores, we later find that the LOO baseline produces scores that are closer to our pLOO $_{improved}$ than the original pLOO.

fore, our pLOO$_{improved}$ and the original pLOO produce similar scores. However, when the model memorized fewer points, as in the case of RESNET-18 on CIFAR-10, the difference becomes starker (Figure 4). This is because, amongst the top 5,000 points, we have a larger distribution of scores, not just the ones with the maximum value.

2. In all cases, the two methods have a non-zero RMSE. This points to the fact that pLOO$_{improved}$ and pLOO produce diverging memorization scores. However, what remains unknown *which one of these two methods produces accurate memorization scores i.e., that are closest to the LOO baseline*.

**Part 2: LOO vs pLOO$_{improved}$ vs pLOO:** To ascertain which one of the two methods is more accurate, we compare pLOO$_{improved}$ and pLOO against the baseline LOO, which is the gold standard for calculating memorization scores. Our results, shown in Figure 5, demonstrate that that the scores from our method have a significantly smaller error than the original pLOO method. For example, 60 of the total 160 points from pLOO$_{improved}$ have 5% error. In stark contrast, *none* of the pLOO meet this criteria. This is a significant finding, and erodes the trust in the original pLOO method. Further more, we found that the RMSE between pLOO and LOO is around 35.50. On the other hand, the RMSE between our improved method and LOO is around 12.19. This means pLOO$_{improved}$ reduces the error by over 65%. This shows that our method produces more accurate scores while requiring 90% fewer shadow models.

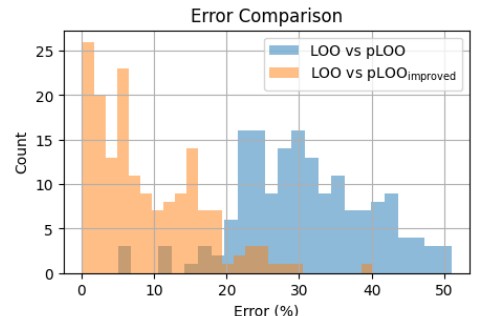

Figure 5: The approximation error between the standard LOO baseline and the two methods. We can see that our pLOO$_{improved}$ has a significantly smaller approximation error than pLOO.

# 6 DISCUSSION AND TAKEAWAYS

**Need to evaluate future approximation methods against the LOO baseline:**

During our study, we compared the memorization scores produced by pLOO and our pLOO$_{improved}$ against the LOO baseline. This type of exhaustive experimental analysis using the LOO baseline is not common in ML literature. As a consequence, we find that one of the most commonly used approximation methods for memorization performs poorly against the standard LOO test. This raises concerns over other techniques that use pLOO in their underlying experimental framework.

Therefore, future researchers need to compare their approximation techniques with the standard benchmarks and ascertain the efficacy of their methods.

**pLOO$_{improved}$ is more accurate because it drops fewer points during sampling:**

However, one question that remains unanswered is why pLOO produces inaccurate results. We believe this is because pLOO drops a large number of points during sampling. For example, in the case of CIFAR-10, pLOO drops $((100 - r)/100) \cdot s$ points. In contrast, pLOO$_{improved}$ samples from a smaller space $s - g$, and drops $((100 - r)/100) \cdot (s - g)$. For example, at a sampling ratio $r$ is 70% (as in the original paper), the number of generalized points $g$ is 45,000, and the total number of points $s$ is 50,000, then pLOO drops approximately $((100 - 70)/100) \cdot 50,000 = 15,000$ points and pLOO$_{improved}$ only drops $((100 - 70)/100) \cdot (50,000 - 45,000) = 1,500$ points per data shard. Dropping fewer points brings the data shard composition closer to the LOO baseline, which only drops a single point at a time. One way to fix this problem in pLOO is to use a much larger sample ratio $r$ value so that fewer points are removed. However, this will require creating even more shards to get a better spread of the data, resulting in even more shadow models, and consequently, a much worse bottleneck. Similarly, one simple way to further reduce the RMSE for our scores from pLOO$_{improved}$ would be to run it for a smaller set of memorized points. For example, instead of creating 200 shards from 5,000 points, generate 200 shards for the top 2,500 points, and another 200 for the next 2,500. However, this requires training more models which might not be possible for every user.

**Our modifications can also improve other existing methods:**

pLOO is the cornerstone for other techniques, such as membership inference attacks, that designed to quantify the leakage of private data in ML models (Salem et al., 2018; Zarifzadeh et al., 2024; Carlini et al., 2022a; Watson et al., 2021; Long et al., 2020; Sablayrolles et al., 2019; Song & Mittal, 2021; Jayaraman et al., 2020; Shokri et al., 2017; Yeom et al., 2018; Tang et al., 2022). However, as discussed earlier, one main drawback of pLOO is inaccurate memorization scores. By improving the scores, we believe it is possible to also improve the membership attacks as well. Specifically, by getting scores precise memorization scores, it should be possible to improve the power of existing membership inference attacks at low false positive rates (i.e., identify vulnerable points without making mistakes), and that too, at a fraction of the original computation cost. Therefore, we believe our work can have a broader impact in the space of ML privacy.

## 7 LIMITATIONS

In this work, we develop an efficient and accurate method of identifying memorized points. However, several limitations must be acknowledged:

**Lack of Theoretical Grounding:** Our method relies heavily on empirical observations and validation. While the proxy (ApB) shows a strong correlation with memorization scores, we do not provide a formal theoretical analysis of its underlying mechanisms. This leaves room for future work to establish a deeper theoretical understanding of why the proxy performs well and how it relates to the broader memorization dynamics.

**Limited Evaluation in Non-Standard Settings:** Our proxy is limited in its applicability to non-standard settings, specifically the continual learning scenario. Here, all points from the most recent tasks might be assigned high memorization probabilities, potentially limiting the method's utility. Since pLOO can still work in this setting, one simple method would be to use pLOO$_{improved}$ to get the raw scores in the continual learning scenario. Future work can aim to extend this proxy to other learning scenarios.

## 8 CONCLUSION

In this work, we develop a method to efficiently identify memorized points and accurately calculate the memorization scores. We show that our improvements significantly reduce the computational requirements and reduce the approximation error. We do this by first identifying the computation bottleneck in the popular pLOO method, i.e., it is run on the entire dataset, and show that most models only memorize a fraction of the data. Next, we develop a simple proxy to identify these

memorized points. We do this by first validating our hypothesis that memorized points are learned towards the end. We use this knowledge to build our proxy, which consists of calculating how early a point is learned during training. Our proxy has a strong relationship (Pearson correlation $< -0.95$) with the memorization scores across all the datasets and models we evaluate. Finally, we incorporate the proxy into the pLOO method by running pLOO only on the points we identified as memorized using our proxy. In doing so, we can reduce the computational cost by over 90% (by requiring training only 200 models, instead of the original 2,000) and improve accuracy by over 65% (by reducing the error from 35.5 to 12.19). As a result, our method provides an effective and accurate tool for identifying memorization that can enable researchers to study memorization for larger more complex models.

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

## A   MODEL ARCHITECTURES:

**VGG6 Architecture:** $64 \rightarrow MaxPool \rightarrow 64 \rightarrow MaxPool \rightarrow 64 \rightarrow MaxPool \rightarrow 64 \rightarrow MaxPool \rightarrow 512 \rightarrow MaxPool \rightarrow FC$.

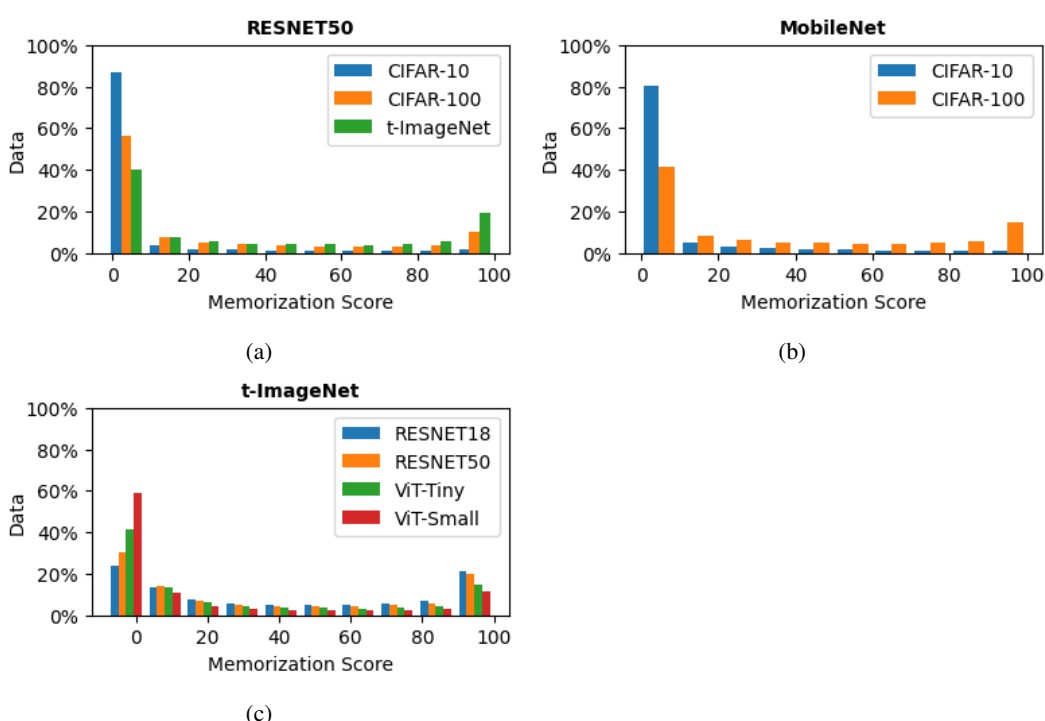

Figure 6: We can see that most of the data is not memorized by the model.

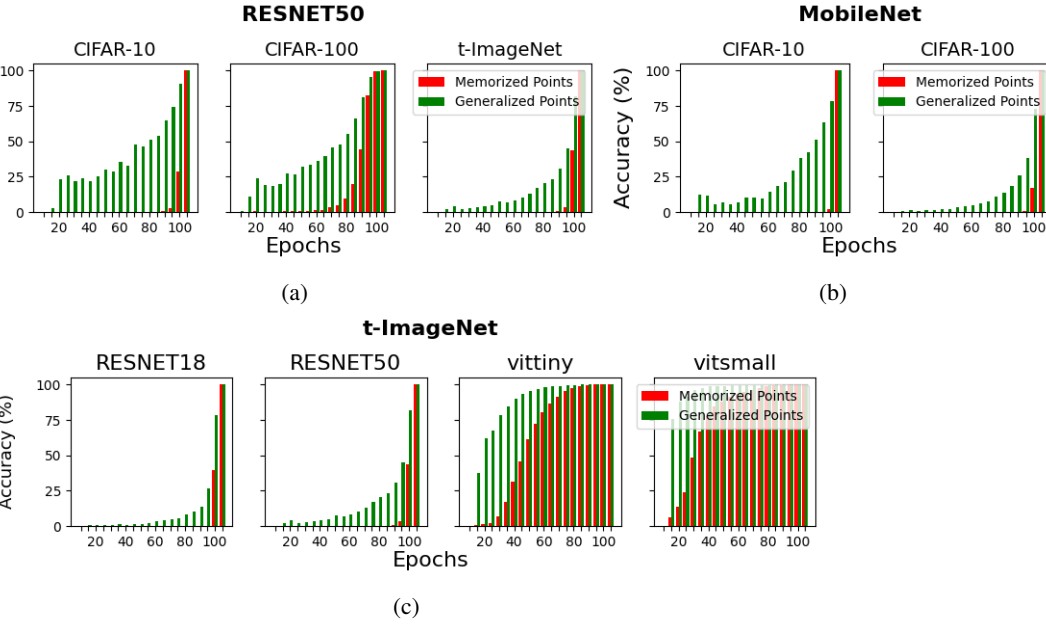

Figure 7: Similar to other models and datasets we evaluate, memorization starts happening towards the end of training.

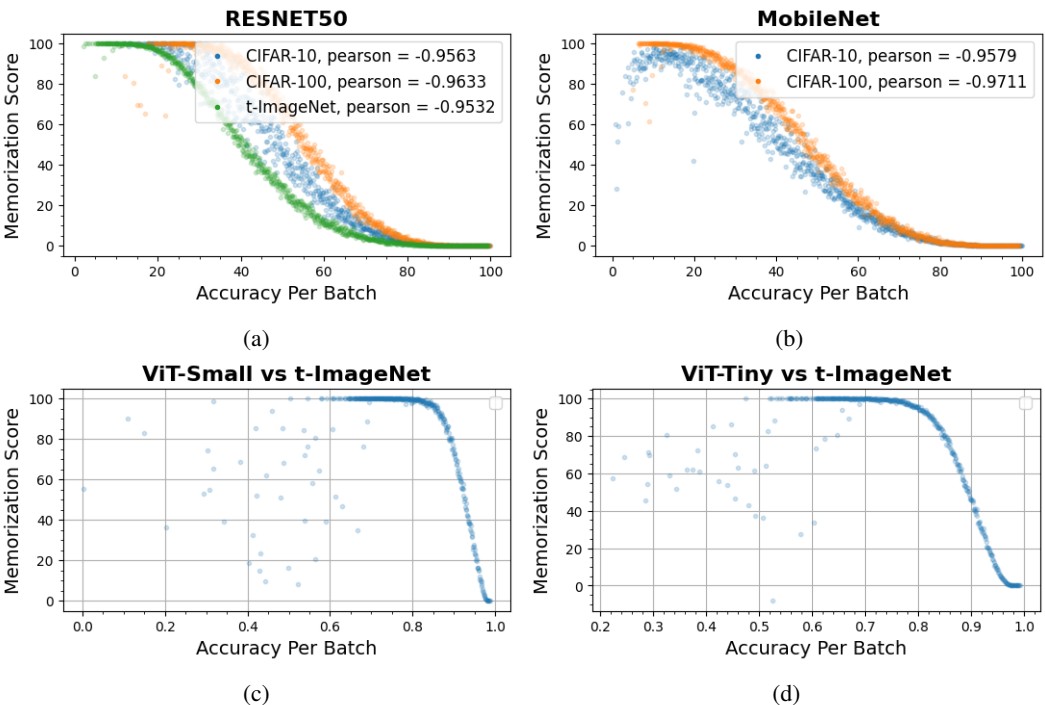

Figure 8: We can see that our proxy has is strongly correlated with the memorization scores. The only outliers here are the ViT models. They memorize and learn faster than other models since we are using a pre-trained baseline.

