# OpenReview forum: "Towards Efficient and Accurate Identification of Memorization in Deep Models"
_ICLR.cc/2025/Conference — Submitted to ICLR 2025_

### Official Review · Reviewer_V7FR · 2024-10-30

**Soundness:** 3
**Presentation:** 2
**Contribution:** 3
**Rating:** 6
**Confidence:** 5

**Summary:**

This paper seeks to understand and detect memorization of deep neural networks. The authors wish to improve upon prior work, that was computationally too expensive to scale to large-scale datasets and models. The key observation is that outliers are most likely to be memorized, as they are not a part of the generalizable portions of the dataset (that is characterized by many similar samples). Further, the outliers that are likely to be memorized are learned late in the training process. Using this idea they are able to detect which samples are most likely to be memorized, thereby reducing the number of samples that need be carefully analyzed for memorization (reducing computational cost). They also show improved results relative to pLOO, which needs to make approximations for computational efficiency, which here are shown to possibly introduce high errors. So the proposed approach is deemed by the authors both computationally faster and also more accurate in detecting memorized training samples.

**Strengths:**

The ideas in this paper are sound, and the experimental results are convincing.

**Weaknesses:**

The main criticisms are on the writing. Generally the paper is written well, but there are places where much improvement is needed. See below for details.

**Questions:**

1. Please define what a "shard" is. The reviewer understands what is meant, but it would be good to define this term, so there is no ambiguity.

2. More important:

- On p. 3, it is written "In short, our method is not only faster than pLOO, but also
more accurate as well. In short, our method is both significantly faster and more accurate than
pLOO, the most popular method for finding memorized points in current literature.
In short, our work makes the following contributions:"
This is three consecutive sentences that begin "In short," which is very poor. And worse, the first and second sentences are essentially duplications.

- On p. 4, "In other words, we train 2000 instances of each of the models (i.e., 1000 models where xi is present
and 1000 models where xi absent from the training data ). 100% models output the ground-truth
label for xi (i.e., all 1000 instances classified the point correctly). However, once xi is removed
from the training data, 25% of the models assign xi the ground-truth label (i.e., 250 out of the 1000
instances classified the point correctly). The resulting memorization score is 100%−25% = 75%."
There is a lot of detail here for a fairly simple concept. Also, I assume you do not always train 2000 instances, but this is an example. I guess better would be, "In other words, *if* we train 2000 instances ..." This is just an example. The writing generally can be made a bit more serious technically.

- Which brings me to the issue of cats. There seem to be a fixation with cats. At the bottom of p. 3 and top of p. 4 there are discussions of cats and outliers. Then again at the middle of p. 4. Cats are also mentioned in the Introduction. This is a minor point, but it distracts the reader, because the discussion becomes somewhat casual, and kinda assumes the reader knows something about the "history" of cats in ML. I suggest trying to use less casual writing.

- At the beginning of Sec. 4.3 you state you have "validated" your hypothesis. I think this is a strong statement. I think you have found *evidence* for your hypothesis, but I am not sure you have validated it.

- In the middle of p. 9 you begin a paragraph "In this work, ..." and then at the beginning of the Conclusions you also begin as "In this work, ...." This is small, but it is indicative of a need to improve the writing in small ways throughout the paper.

---

> ### Author Response · Authors · 2024-11-15
>
> We thank the reviewer for their feedback. We have incorporated all their suggestions into our work. Most of the comments seemed to be related to textual changes. Regarding your points 2a) details for simple concepts and 2b) using cat examples, we tried to provide intuition so the work is digestible to readers unfamiliar with this space. However, we did modify the text to reflect your feedback. If there is anything else we can do to improve our work, please let us know and we would be happy to do so.

---

> > ### Author Response · Authors · 2024-11-25
> >
> > Dear Reviewer V7FR,
> > Thank you for your valuable feedback. We have provided detailed responses to your questions and hope they address your concerns thoroughly. As the interactive discussion period is coming to a close, please don’t hesitate to reach out if you need further clarification or additional information.

---

> > > ### Author Response · Authors · 2024-11-30
> > >
> > > Dear Reviewer,
> > >
> > > As we approach the discussion deadline, please let do not hesitate to let us know if you have any more questions or suggestions.

---

### Official Review · Reviewer_xh5N · 2024-10-31

**Soundness:** 2
**Presentation:** 2
**Contribution:** 2
**Rating:** 3
**Confidence:** 4

**Summary:**

This paper studies the memorization ability of neural networks by proposing an efficient approximate method to identify memorized points and consequently calculate the memory score. The development is based on the observation that memorized points are learned towards the end of the training cycle. Imperical results indicate that the proposed method can accurately predict the memorization scores by reducing the computational overhead by over 90%.

**Strengths:**

The paper focus on a sub-problem in the pLOO paper, which is to define efficient method to estimate the memorization ability of neural networks. The paper is in general written clearly. The problem of estimating the memorization ability of neural network is important to understand some properties of neural networks. The proposed method is intuitive and efficient.

**Weaknesses:**

I feel the paper in general is not complete. The current version explains the problem and proposes an efficient solution based on an intuition of the training dynamic of neural networks. Although the explanation is clear, most of the definitions and problem settings have been proposed by previous works. I think there are more work to be done to make the paper more complete. For example, more experiments need to be conducted to verify the effectiveness and usefulness of the proposed method. The authors should at least follow the experimental setting in the pLOO paper to make the experimental part richer and more extensive. The current experiments are too simple to drawn conclusions from.

Furthermore, if possible, a theoretical explanation on the proposed method would make the paper more solid.

**Questions:**

Please refer to the weaknesses part.

---

> ### Author Response · Authors · 2024-11-15
>
> >"The authors should at least follow the experimental setting in the pLOO paper to make the experimental part richer and more extensive."
>
> We thank the reviewer for their comments and recognizing our solution is far more extensive than the existing SOTA. However, their comment regarding our experimental setting, as being less extensive than the original pLOO, is not correct. In fact, we conduct a more extensive evaluation than the original work. 1) The original paper only evaluated a single model (Resnet) using a single architecture. In contrast, we evaluate three different families of models (MobileNet, VGG, and Resnet). 2) The original work did not evaluate their score accuracy against the LOO baseline (gold standard). In contrast, we do and show that our scores are more accurate as well.
> If there is any specific experiment the reviewer has in mind, please let us know.
>
>
>
>
> >"Furthermore, if possible, a theoretical explanation on the proposed method would make the paper more solid."
>
> The reviewer is right to point out the value of a theoretical explanation. Though our work focused on empirical validation and practical utility rather than theoretical derivation, we do provide an intuition behind our method in Section 2. We will explicitly mention the potential for future theoretical work to analyze our proxy in the limitations section.

---

> > ### Author Response · Authors · 2024-11-25
> >
> > Dear Reviewer xh5N,
> > Thank you for your valuable feedback. We have provided detailed responses to your questions and hope they address your concerns thoroughly. As the interactive discussion period is coming to a close, please don’t hesitate to reach out if you need further clarification or additional information.

---

> > > ### Comment · Reviewer_xh5N · 2024-11-26
> > >
> > > Thanks for the rebuttal. However, I cannot agree with the response:
> > >
> > > 1.  The original paper only evaluated a single model (Resnet) using a single architecture. In contrast, we evaluate three different families of models (MobileNet, VGG, and Resnet): the original paper has actually considered ResNet50, ResNet18, Inception and DenseNet100 (see Figure 7).
> > > 2. Our solution is far more extensive than the existing SOTA: In the pLOO paper, the authors have conducted experiments to examine many aspects of the proposed method, see Section 3.1 -- 3.6. However, this paper only compares the memorization scores. I am not convinced this is more extensive.
> > >
> > > Thus, I will maintain my current score.

---

> ### Author Response · Authors · 2024-11-27
>
> Thank you for the feedback.
> > Figure 7
>
> Apologies for our oversight regarding Figure 7. To verify the results of our study, we ran additional experiments (placed in the appendix) across more models (Resnet50, ViT-Small, Vit-Tiny). At this point we have evaluated our models against:
> - different architectures (MobileNet, VGG, ResNet, ViT)
> - different models complexity (by varying the depth of the same models e.g., ResNet18 and 50, and ViT Tiny and Small)
> - different datasets (Cifar10,100, and t-imagenet)
> - trained from scratch (Mobilenet, VGG, ResNet) and with pre-trained base models (ViT)
>
> This results in 12 data points that confirm our findings. If the reviewer feels we missed any particular set of training parameters, please do not hesitate to let us know.
>
> > In the pLOO paper, the authors have conducted experiments to examine many aspects of the proposed method,
>
> Our evaluations consist of observing how models learn/memorize *natural* data. In contrast, the reviewer cited paper has focused on artificial memorization (via noisy images or randomized labels). There is a possibility that two might behave differently [1]. Therefore, part of our contribution is to verify the existence of this behaviour in natural points. However, if there is any particular evaluation that the reviewer would be interested in, please let us know.
>
> [1] Evaluations of Machine Learning Privacy Defenses are Misleading

---

> > ### Author Response · Authors · 2024-11-30
> >
> > Dear Reviewer,
> >
> > As we approach the discussion deadline, please let do not hesitate to let us know if you have any more questions or suggestions.

---

### Official Review · Reviewer_thK4 · 2024-11-02

**Soundness:** 3
**Presentation:** 2
**Contribution:** 2
**Rating:** 5
**Confidence:** 4

**Summary:**

To mitigate the computation cost of identifying memorized points in deep neural networks, this paper proposes a heuristic meric, ApB, to efficiently identify memorized points and accurately calculate the memorization scores. With incorporating pLOO method by running pLOO only on the points filtered by ApB, empirical experiments show that the proposed method, compared with pLOO, can reduce the computational cost by over 90%  and improve accuracy by over 65%.

**Strengths:**

1.	The presentation is easy to follow and understand.

2.	The paper propose to reduce computing cost of identifying memorized points by only evaluating a fraction of samples, filtered by ApB, which is straightforward, efficient and effective.

**Weaknesses:**

1.	In this paper, the motivation of studying identifying memorized points and related works are not well presented, making it difficult to understand the position of the paper in the literature and the impact of the paper. Some more details are needed, like why identification of memorization is important and some more works related to identifying memorization.
2.	The main hypothesis studied in this paper, “Generalized points are learned faster than memorized ones during training”, was extensively studied and verified in the literature, like [1][2]. The only contribution of this paper is introducing a heuristic metric ApB to filter some points to identify memorized points.

[1] Stephenson, Cory, et al. "On the geometry of generalization and memorization in deep neural networks." International Conference on Learning Representations.
[2] Gu, Jindong, and Volker Tresp. "Neural network memorization dissection." arXiv preprint arXiv:1911.09537 (2019).

**Questions:**

1.	The definition of the metric ApB is not clear. It’s better to define it mathematically.  To my understanding, after every epoch, it does inference for every examples in the training dataset, then calculate the ration of being correct for each samples over all the epochs. Please help confirm if my understanding is correct?
2.	In section 4.1, what method is utilized to obtain memorization scores? LOO? pLOO? And in section 4.2, how the labels are obtained, for generalized points and memorized points?
3.	In figure 1, that “CIFAR-10 has consistently fewer memorized points across all models compared to CIFAR-100” may be because each class in CIFAR10 has more samples(5k) than CIFAR100(500), since it’s known that classes with fewer samples will lead to more memorized points, such as the example in line 193.
4.	Almost all the figures are overwide and need to be re-organized. Moreover, please refer to template instructions to use \citet and \citep appropriately.

---

> ### Author Response · Authors · 2024-11-15
>
> > ...why identification of memorization is important and some more works related to identifying memorization.
>
> We thank the reviewer for this comment. We can elaborate on the necessity for finding memorization points. These range from detecting privacy leakage, fairness, data mislabeling [1,2] etc. However, the current pLOO method makes finding these points very computationally expensive. Our goal is to reduce these costs to help alleviate the aforementioned issues. Similarly, regarding more works related to identifying memorized points, Feldman et al is the most popular means of doing so in current literature. Therefore, we compare our technique against the SOTA.
>
> >The main hypothesis studied in this paper, “Generalized points are learned faster than memorized ones during training”, was extensively studied and verified in the literature, like [1][2]. The only contribution of this paper is introducing a heuristic metric ApB to filter some points to identify memorized points.
>
> We thank the reviewer for sharing these works. However, both papers study artificial memorization i.e., introducing new points to a class distribution (either through mislabeling or using noisy points). None of them study natural memorization, i.e., memorization of the actual points in the dataset. As we know from recent work, artificial memorization might not always extend to natural points [1]. Therefore, our first contribution is to extend this observation to real points. Second, ML practitioners are more concerned with what points the model memorized from their datasets instead of artificially introduced ones. Our technique helps bridge that gap. Third, our contribution goes beyond just the ApB proxy. This is because we also developed a method to incorporate that proxy into the existing pLOO method. In stark contrast, neither of two papers develop a proxy for natural points or even address how to improve the pLOO method for any real-world application.
>
>
> >The definition of the metric ApB is not clear...
>
> The intuition is correct. We have updated the paper to include the precise algorithm. We thank the reviewer for this comment.
>
> >In section 4.1, what method is utilized to obtain memorization scores? LOO? pLOO? And in section 4.2, how the labels are obtained, for generalized points and memorized points?
>
> In 4.1, we use pLOO to verify our initial hypothesis that the models generalize more than memorize. In 4.2, we compare the ground truth label to the model prediction. We would be happy to clarify if the reviewer has any other questions.
>
> >In figure 1, that “CIFAR-10 has consistently fewer memorized points across all models compared to CIFAR-100” may be because each class in CIFAR10 has more samples(5k) than CIFAR100(500), since it’s known that classes with fewer samples will lead to more memorized points, such as the example in line 193.
>
> We thank the reviewer for this comment. Based on our current understanding of memorization, this observation is correct, and is likely a product of the resulting train-test gap, as we mention in Section 4.1.
>
> [1] Memorisation in Machine Learning: A Survey of Results
>
> [2] Leave-one-out unfairness

---

> > ### Author Response · Authors · 2024-11-25
> >
> > Dear Reviewer thK4,
> > Thank you for your valuable feedback. We have provided detailed responses to your questions and hope they address your concerns thoroughly. As the interactive discussion period is coming to a close, please don’t hesitate to reach out if you need further clarification or additional information.

---

> ### Comment · Reviewer_thK4 · 2024-11-26
>
> Thank you to the authors for providing the detailed responses. Some of my concerns have been addressed. But still, the writing is one of the major weaknesses of this paper.  Although the author already improved the paper with the reviewers' suggestions, some confusion still exists. For example,
> 1. in *"Line 320, we calculate the ApB over the last batch of each epoch"*, this doesn't match the algorithm 1 presented. Which one is the real implementation?
> 2. In figure2, what does the y-axis denote? I guess it's accuracy?
> 3.  *"In 4.2, we compare the ground truth label to the model prediction"*.  Where is the ground truth label from? Is it calculated by LOO?
>
> I suggest the author thoroughly proofreads the paper to make the important information clear and reduce confusion. Besides, I suggest the author incorporate the necessity of finding memorization points in the introduction to help readers recognize the impact and importance of the research topic the paper studies.
>
> Besides, I agree with the reviewer xh5N that the contribution of this work is not significant. From the experiments in the paper,  we can see the proposed heuristic method ApB, further incorporated with pLOO, is more efficient and effective compared with pLOO, on cifar10, cifar100, t-ImageNet datasets. To enrich the value of this work, I'd expect some insights from the paper for memorization identification problems (the main hypothesis for ApB was studied in the literature before) or theoretical grounding to support it's generalization capability to other datasets. As such, I will keep my current scores.

---

> ### Author Response · Authors · 2024-11-27
>
> Thank you for the feedback. We will clarify the text in the paper.
>
> > Lack of Theoretical Grounding
>
> To verify the results of our study, we ran additional experiments (placed in the appendix) across more models (Resnet50, ViT-Small, Vit-Tiny). At this point we have evaluated our models against:
> - different architectures (MobileNet, VGG, ResNet, ViT)
> - different models complexity (by varying the depth of the same models e.g., ResNet18 and 50, and ViT Tiny and Small)
> - different datasets (Cifar10,100, and t-imagenet)
> - trained from scratch (Mobilenet, VGG, ResNet) and with pre-trained base models (ViT)
>
> This results in 12 data points that confirm our findings. If the reviewer feels we missed any particular set of training parameters, please do not hesitate to let us know.
>
> > The main hypothesis for ApB was studied in the literature before
>
> Our evaluations consist of observing how models learn/memorize *natural* data. Most literature has focused on artificial memorization (via noisy images or randomized labels). The two sets of points can behave differently, leading to inconsistent findings [1]. Part of our contribution is verifying, that in this case, the behavior is consistent. However, if there is any particular evaluation that the reviewer would be interested in, please let us know.
>
> "Evaluations of Machine Learning Privacy Defenses are Misleading," Michael Aerni, Jie Zhang, Florian Tramèr, CCS 2024

---

> > ### Author Response · Authors · 2024-11-30
> >
> > Dear Reviewer,
> >
> > As we approach the discussion deadline, please let do not hesitate to let us know if you have any more questions or suggestions.

---

### Official Review · Reviewer_2AeM · 2024-11-04

**Soundness:** 2
**Presentation:** 2
**Contribution:** 2
**Rating:** 6
**Confidence:** 4

**Summary:**

The paper addresses computational and accuracy limitations in measuring memorization through the "pseudo-leave-one-out" (pLOO) (Feldman & Zhang, 2020). The key idea is to leverage an insight that data points that are memorized tend to be misclassified in early training stages and only correctly classified in later stages. Based on this observation, the authors propose a proxy metric termed Accuracy per Batch (ApB) to identify candidate memorized training points. (This could also be used as a heuristic to detect memorized points.) Using these proxy scores, they develop a more efficient sampling strategy for pLOO, achieving a 90% reduction in computational cost and a 65% reduction in error.

**Strengths:**

- The paper is well-written, has clear motivation, and presents valuable contributions to the ICLR community.
- The insights are novel and can be practically significant, with the proposed ApB metric demonstrating improved efficiency and accuracy compared to pLOO.

**Weaknesses:**

- Several writings in the paper lack technical justification. For example, the intuition presented in Section 2 lacks technical rigor. While the authors demonstrate convincing empirical results on image classification tasks to support their claims, its applicability or generability to other domains remains uncertain. It would be useful to extend the analysis (e.g., memorized data points have a high loss at the early stage of training and a low loss at the later stage) on simpler models such as linear regression to gain technical insights. Although I believe that extensive theoretical analysis is not required to meet the bar for ICLR presentation, this is certainly one weakness of the paper.
- The authors’ claim that the memorized points constitute a small fraction of training data seems to be overly specific to certain datasets and architectures. In extreme cases, such as when all labels are randomized (e.g., [1]), the model must memorize all points to obtain 100% training accuracy. I believe that the authors’ claim can be generally true for academic datasets with no label noise but might not hold in more practical (or broader) settings.
- The limitations of the proposed approach are not sufficiently discussed. While pLOO offers general applicability across various settings (e.g., we can train the models on various subsets of the dataset), including more non-typical settings like continual learning scenarios, it's unclear how ApB can be extended to non-standard settings. For example, ApB will assign a high memorization probability to all data points that appeared in the last task.
- Another concern is that the authors treat noisy LOO estimates as ground truth. LOO has known reliability issues in machine learning [2], which may affect the conclusion and results of the paper. The pLOO provides guarantees based on the number of models trained (in the infinite limit), suggesting that a more appropriate baseline would be pLOO with an increased number of retraining rather than relying on potentially unreliable baseline measures. In addition, ApB does not offer such a guarantee since it produces a bias in the data selection process, which can be another limitation.
- The experimental setup lacks sufficient detail for reproducibility (e.g., learning rate and batch size used).

**Questions:**

(I’ve included additional suggestions as well as the questions.)
- I feel like several claims in the paper are overstated. For example, line 58's claim about requiring "hundreds of thousands of models, computationally intractable" contradicts existing literature showing this has been done on CIFAR-10 (e.g., [3]). (It is technically not intractable.) Similarly, line 230's comment about pLOO's inapplicability to production models raises questions about the method's broader applicability for obtaining memorization scores.
- The examples in lines 157-161 appear redundant and could be omitted.
- Could the authors provide a precise definition of ApB in Section 4.3?
- Do the authors think the insights presented in this paper hold for other settings (e.g., regression or language models)?
- (Minor) There are several instances of incorrect usage of \citet and \citep.
- (Minor) The phrase "in short" is repeated in line 113.
- (Minor) Line 241 should say "a small fraction of the dataset" rather than just "a fraction."
- (Minor) Section(5) contains formatting issues at line 377.

[1] What Do Neural Networks Learn When Trained With Random Labels? [2] A Bayesian Approach To Analysing Training Data Attribution In Deep Learning [3] Datamodels: Predicting Predictions from Training Data

---

> ### Author Response · Authors · 2024-11-15
>
> >Several writings in the paper lack technical justification…
>
> The reviewer is right to point out the value of a theoretical justification. Though our work focused on empirical validation and practical utility rather than theoretical derivation, we do provide an intuition behind our method in Section 2. We will explicitly mention the potential for future theoretical work to analyze our proxy in the limitations section.
>
> Similarly, we appreciate the comment about extending memorization to linear regression or LLM.  However, the Feldman et al definition only extends to classification models. LLMs have a variety of definitions of memorization [1], due to their auto-regressive nature. While it might be possible to extend our intuition to other domains, that is out of the scope of the current work.
>
> [1] https://arxiv.org/pdf/2310.18362
>
>
> >The authors’ claim that the memorized points constitute a small fraction of training data seems…
>
> This is a good observation. And we agree, that small fraction of memorization is not possible in _all_ cases. But in general, memorization should be less than generalization for a real-world models. This is because real-world models are designed to have the smallest train-test gap, and we know from current literature, as the gap decreases, memorization also decreases [2]. In a time when we see models are surpassing human performance on a variety of tasks, at times, near perfect accuracy, it is expected that these models will memorize less than they generalize.
>
> Furthermore, even in the extreme case where the models memorize most of the data, running the vanilla pLOO algorithm is still computationally wasteful. As we do not want to spend compute calculating scores of points that the model generalized. We will be happy to include this nuance in the text.
>
> [2] Membership Inference Attacks From First Principles
>
> >The limitations of the proposed approach are not sufficiently discussed…
>
> This is an excellent point by the reviewer. It is correct that during continual learning, points from the last task will have a higher memorization probability. Since our algorithm provides _relative_ scores, i.e., which points are memorized with the highest score _within_ a given task. For example, consider a model that is pre-trained on the Imagenet dataset, and then fine-tuned on MRI dataset (i.e., enables intra-dataset comparison). Our algorithm can help find which points from the MRI dataset are most likely memorized during fine tuning. However, it is true, as the reviewer points out, that it does not allow us to aggregate the two datasets and compare the memorization approximations between them (i.e., inter-dataset comparison). We will be happy to clarify this in the limitations section. At the same time, it is important to recognize that pLOO method will not work in the continual learning scenarios either. So it is a limitation of both methods.
>
> >Another concern is that the authors treat noisy LOO estimates as ground truth.
>
> The author is right to point out that, under small repeats, the LOO baseline is noisy for **train-test influence** estimation (as pointed out in the reviewer cited paper). However, this critique does not apply to our work for a few reasons:
> 1. Even the authors of the original pLOO accept that the LOO method is the gold standard and that pLOO is an attempt to approximate LOO, not the other way around.
>
> 2. Our work is focused on self-influence, instead of the train-test influence (which is the focus of the reviewer’s citation). LOO for self-influence is less noisy as it is ascertaining whether a point is classified correctly if missing from the data. And outliers are least likely to be classified correctly.
>
> 3. Noise reduction is precisely why we repeat our LOO experiments dozens of times (Section 5), as suggested in the original pLOO work. This reduces the noise for LOO providing a more exact approximation.
>
> 4. Lastly, pLOO has one more source of noise than LOO, the removal of corresponding subpopulations. For example, if there are data shards with a subpopulation missing, the error in the pLOO calculation is exacerbated (Section 5).
>
> >The experimental setup lacks sufficient detail for reproducibility (e.g., learning rate and batch size used).
>
> We provide these details in para 2 Section 4.
>
> >Minor Suggestions:
>
> We thank the reviewer for the remaining suggestions. We have modified the text to reflect them.

---

> > ### Author Response · Authors · 2024-11-25
> >
> > Dear Reviewer 2AeM,
> > Thank you for your valuable feedback. We have provided detailed responses to your questions and hope they address your concerns thoroughly. As the interactive discussion period is coming to a close, please don’t hesitate to reach out if you need further clarification or additional information.

---

> > > ### Comment · Reviewer_2AeM · 2024-11-26
> > >
> > > I thank the authors for their reply. I acknowledge that I have read the authors' responses and other reviewers' comments. I somewhat agree with reviewer xh5N that the original paper had a more in-depth analysis (with experiments on several architectures). Still, I understand that the scope of this paper is to build a better tool for detecting memorized data points rather than subsequent analysis with memorized data points. (Although it would be useful to show how more correct detection can lead to more interesting analysis).
> > >
> > > > This is an excellent point by the reviewer. It is correct that during continual learning, points from the last task will have a higher memorization probability. Since our algorithm provides relative scores, i.e., which points are memorized with the highest score within a given task. For example, consider a model that is pre-trained on the Imagenet dataset, and then fine-tuned on MRI dataset (i.e., enables intra-dataset comparison). Our algorithm can help find which points from the MRI dataset are most likely memorized during fine tuning. However, it is true, as the reviewer points out, that it does not allow us to aggregate the two datasets and compare the memorization approximations between them (i.e., inter-dataset comparison). We will be happy to clarify this in the limitations section. At the same time, it is important to recognize that pLOO method will not work in the continual learning scenarios either. So it is a limitation of both methods.
> > >
> > > Could the authors elaborate on why pLOO wouldn't work in this setup? If we were to analyze the effect of memorization in the pre-trained datasets after fine-tuning, couldn't we still apply the same retraining strategies (leave some data points out in the pre-trained stage)? As the reviewer comments, briefly mentioning these limitations would be helpful.
> > >
> > > Since the authors have clarified most of my points, I have increased my score to 6.

---

> ### Author Response · Authors · 2024-11-28
>
> > Original Paper had experiments on several architectures
>
> To address this concern, we ran additional experiments (placed in the appendix) across more models (Resnet50, ViT-Small, Vit-Tiny). At this point we have evaluated our models against:
> - different architectures (MobileNet, VGG, ResNet, ViT)
> - different models complexity (by varying the depth of the same models e.g., ResNet18 and 50, and ViT Tiny and Small)
> - different datasets (Cifar10,100, and t-imagenet)
> - trained from scratch (Mobilenet, VGG, ResNet) and with pre-trained base models (ViT)
>
> This results in 12 data points that confirm our findings. If the reviewer feels we missed any particular set of training parameters, please do not hesitate to let us know.
>
> > Could the authors elaborate on why pLOO wouldn't work in this setup?
>
> Actually, the reviewer is right. Consider two scenarios: SC-1) a pre-trained base model, which is fine-tuned on tiny-ImageNet and SC-2) training on tiny-ImageNet from scratch. The memorization scores of the same points will be different across the two scenarios. This is because for SC-1, using a pre-train base will allow the model converge faster, leading to a higher accuracy and far fewer memorized points. In contrast, in SC-2 a model trained from scratch will have a lower accuracy and far more memorized points. We ran a very similar experiment by fine-tuning ViT models on Tiny Imagenet (Figure 6 (c)). This means we can compare the memorization scores. However, the ApB scores will be lower for the pre-trained model but higher for model trained from scratch. We have modify the text to show that the limitation in continual learning scenario only applies to ApB. However, since pLOO gives the raw scores, we can use pLOO_improved to get actual memorized scores without having to run pLOO over all the points.

---

> > ### Author Response · Authors · 2024-11-30
> >
> > Dear Reviewer,
> >
> > As we approach the discussion deadline, please let do not hesitate to let us know if you have any more questions or suggestions.

---

### Author Response · Authors · 2024-11-25
**Feedback**

Dear Reviewers,
Thank you for your review.
We have modified the paper to reflect your feedback.
Please let us know if we you have more suggestions on how we can improve our paper.

---

### Meta-Review · Area_Chair_Q6MY · 2024-12-21

**Metareview:**

The submission addresses the computational inefficiency of measuring memorization in deep neural networks using pseudo-leave-one-out (pLOO) methods. The authors propose a heuristic metric, Accuracy per Batch (ApB), as a proxy to identify memorized data points, enabling a more efficient sampling strategy for pLOO. The results indicate that this method can reduce computational cost. Reviewers were concerned about the incremental nature of the work, the practical utility of identifying memorized points, theoretical grounding, clarity, and experimental breadth. The authors were able to partially address some of the concerns (experimental breath) through experiments but issues remain, particularly regarding theoretical grounding and clarity. Some of the concerns of the reviewers like practical utility and terminology (e.g. shards) were not addressed in the manuscript. The manuscript can benefit from additional revisions and resubmission.

**Additional Comments On Reviewer Discussion:**

The reviewer discussion highlighted some issues with the claims of the paper regarding the experimental breadth as compared to pLOO, the reviewers added additional experiments. There were also significant concerns about the clarity, some of them were addressed in line but not all changes propagated to the manuscript.  It was pointed out the main hypothesis of the paper has been studied in related contexts of synthetic memorization. A recurring concern was a lack of theoretical grounding.

---

### Decision · Program_Chairs · 2025-01-22

Reject